# Two-variable nullcline analysis of ionic general equilibrium predicts calcium homeostasis in ventricular myocytes

David Conesa[1☯], Blas Echebarria[1], Angelina Peñaranda[1], Inmaculada R. Cantalapiedra[1], Yohannes Shiferaw[2], Enrique Alvarez-Lacalle[1☯]*

**1** Departament de Física. Universitat Politècnica de Catalunya-BarcelonaTech, Barcelona. Spain, **2** Physics Department. California State University Northridge, Los Angeles, California, United States of America

☯ These authors contributed equally to this work.
* enric.alvarez@upc.edu

**Data Availability Statement:** All relevant data are within the manuscript and its Supporting Information files.

## Abstract

Ventricular contraction is roughly proportional to the amount of calcium released from the Sarcoplasmic Reticulum (SR) during systole. While it is rather straightforward to measure calcium levels and contractibility under different physiological conditions, the complexity of calcium handling during systole and diastole has made the prediction of its release at steady state impossible. Here we approach the problem analyzing the evolution of intracellular and extracellular calcium fluxes during a single beat which is away from homeostatic balance. Using an in-silico subcellular model of rabbit ventricular myocyte, we show that the high dimensional nonlinear problem of finding the steady state can be reduced to a two-variable general equilibrium condition where pre-systolic calcium level in the cytosol and in the SR must fulfill simultaneously two different equalities. This renders calcium homeostasis as a problem that can be studied in terms of its equilibrium structure, leading to precise predictions of steady state from single-beat measurements. We show how changes in ion channels modify the general equilibrium, as shocks would do in general equilibrium macroeconomic models. This allows us to predict when an enhanced entrance of calcium in the cell reduces its contractibility and explain why SERCA gene therapy, a change in calcium handling to treat heart failure, might fail to improve contraction even when it successfully increases SERCA expression.

## Author summary

Cardiomyocytes, upon voltage excitation, release calcium, which leads to cell contraction. However, under some pathological conditions, calcium handling is impaired. Recently, SERCA gene therapy, whose aim is to improve $Ca^{2+}$ sequestration by the Sarcoplasmic Reticulum (SR), has failed to improve the prognosis of patients with Heart Failure. This, together with recent counterintuitive results in calcium handling, has highlighted the need for a framework to understand calcium homeostasis across species and pathologies. We show here that the proper framework is a general equilibrium approach of two

**Funding:** E.A-L, B.E, A. P, I.R.C and D. C received grants from Fundació La Marató de TV3 in 2014, under grant number 20151110, URL:http://aquas.gencat.cat/web/.content/minisite/aquas/ambits/recerca/convocatories_ex_ante/la_marato_tv3/maratotv3_2014_cor.pdf They also received grants from the Spanish Ministerio de Economía, Industria y Competitividad (MINECO) under grant number SAF2017-88019-C3-2-R and later A.P received grant. rom Ministerio de Ciencia Innovación y Universidades grant PGC2018-095456-B-I00. URL: https://sede.micinn.gob.es/stfls/eSede/Ficheros/2017/Propuesta_Resolucion_Provisional_Proyectos_Retos_2017.pdf URL: https://sede.micinn.gob.es/stfls/eSede/Ficheros/2019/Segunda_Propuesta_Resolucion_Definitiva_PGC_2018.pdf Y.S. received grant R01-119095 from National Heart, Lung and Blood Institute (NHLBI) URL: https://www.nhlbi.nih.gov/ Y.S.; B. E., and E.A-L acknowledge also that part of this work was completed in the Kavli Institute of Theoretical Physics which sponsored the summer program Integrative Cardiac Dynamics under grants NSF PHY-1748958, NIH- R25GM067110, and by the Gordon and Betty Moore Foundation (Grant No. 2919.01) https://www.kitp.ucsb.edu/activities/cardio18 The funders had no role in study design, data collection and analysis, decision to publish, or preparation of the manuscript.

**Competing interests:** The authors have declared that no competing interests exist.

independent variables. The development of this framework allows us to find a possible mechanism for the failure of SERCA gene therapy even when it manages to increase Ca SERCA expression.

## Introduction

Complex systems often present robust regulation that makes them resilient to changes in environmental conditions. Despite the presence of a large number of nonlinearly interacting elements, averaged quantities in these systems often attain predictable, and sometimes constant, values. Examples range from ecosystems and climate regulation to long-run macroeconomics [1, 2]. Our bodies present a good example of such behavior. The maintenance of homeostatic equilibrium in temperature, pH, osmolality or ionic concentration is crucial to sustaining life, and complex feedback mechanisms have been developed to achieve such goal.

Another important example is the regulation of heart function. Fundamental for survival is the ability of the heart to contract and expel more blood by increasing the heart rate [3]. In most animal species the amount of blood pumped at each beat increases rather strongly with beat rate, which stems from a positive correlation between contractile force and beating frequency [4–6]. However, the variability of the force-frequency relation is quite large across species, being quite flat for some, like mouse and rat [7, 8]. As the contraction machinery in the sarcomeres is activated by an increase in intracellular calcium, this variability is determined by differences in calcium handling [9–11]. During the action potential of the heart, calcium enters into the cell via L-type calcium channels (LCC) triggering the release of calcium stored in the sarcoplasmic reticulum (SR). Each time voltage raises, the intake of calcium due to the opening of LCC raises the probability of RyR2 to open, releasing calcium from the SR during the depolarization of voltage (systole)—the well-known Calcium-Induced Calcium-release (CICR) mechanism—which then binds to TnC, triggering contraction. Once the voltage is repolarized, calcium released from the SR is uptaken by SERCA back to the SR while the sodium-calcium exchanger (NCX) is able to extrude the calcium which entered via LCC, and contraction ceases. This process seems universal across species, although quantitatively can be very different. Not just the release of calcium is species-dependent, its reuptake into the SR by the SR $Ca^2$ ATPase (SERCA) and the extrusion of calcium from the cell by the sodium-calcium exchanger also differs. Translation of results obtained in animal models targeting different regulatory pathways into human patients is thus not trivial.

To reach homeostatic equilibrium, the amount of calcium entering the cell must be compensated by the quantity extruded [12, 13]. Equally, the amount of calcium released from the SR during the transient must be the same reuptaken by SERCA [12, 14]. Global equilibrium is thus determined by a balance of global fluxes, that encompass the complex internal machinery of calcium release, composed of thousands of interacting release units. A good understanding of calcium homeostasis is necessary to understand the effect of changes in calcium handling regulatory mechanisms. In particular, dysregulations in calcium homeostasis have important implications for arrhythmias since calcium overload of the SR often facilitates the initiation of calcium alternans, calcium waves, and arrhythmogenic EADs [13, 15–17].

Altered expression and/or functionality of proteins involved in calcium dysregulation, has often been treated with therapeutic strategies such as gene therapy [18]. In particular, SERCA gene therapy has been successfully applied to preclinical treatment of heart failure (HF) in animals, showing an increase in SERCA2a expression and improved cardiac function [19, 20]. In humans, despite early promising results [21, 22], more recent studies with wider samples of

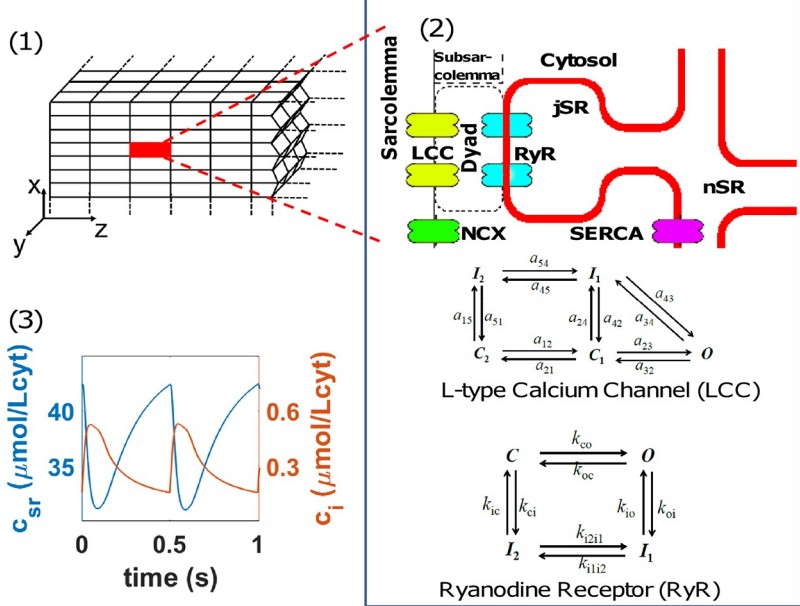

**Fig 1. Structure of the computational ventricular rabbit cardiomyocyte model.** Panel (1): Scheme of the three-dimensional model where the cardiomyocyte is divided into thousands of 3d stacked Calcium Release Units (highlighted one in red). Panel (2): Depiction of the basic structure of the CaRUs showing the L-type Calcium Channels (LCC) in the vicinity of the Ryanodine Receptor (RyR), the SERCA pump and the Na-Ca Exchanger. The different states of the LCC and the RyR2 are also depicted. Panel (3): Typical cytosolic (orange) and SR (blue) free calcium transients in the rabbit ventricular model (described in Methods) at a voltage-clamped pacing rate of 2 Hz.

patients have found no statistical effects of the treatment [23]. These unsuccessful results have been related to reduced viral gene transfer in humans and the effect of neutralizing antibodies present in patients. However, one can not disregard the possibility that SERCA therapy may fail even when SERCA function is improved since its failure or success also depends on how the underlying pathology affects the general equilibrium properties of calcium homeostasis. Highlighting also the relevance of understanding calcium homeostasis in order to predict calcium levels when some channel properties are changed, recent reviews [12, 24, 25] have shown clear counterintuitive calcium behavior. For example, an increase in entrance of calcium via LCC is not always followed by an increase in total calcium levels at the steady state.

To clarify why the calcium behavior seems often counterintuitive and show its very complex nature we can use that precise example. Consider that, in a cell under constant pacing, the conductance of the L-type calcium channels is suddenly increased. Naively, one would expect this to lead to an increase in SR calcium load, as more calcium enters the cell, and gets accumulated into the SR. This is indeed often the case, as our simulations show for a detailed model of calcium handling in rabbit myocytes (see Fig 1 and details of the model in the next section). It is not too difficult, however, to modify the model with changes in the levels of buffers and SERCA, and obtain counterintuitive results. In this modified model, as the strength of LCC increases, the SR Ca concentration decreases as seen in Fig 2. In this case, a larger entrance of calcium into the cell produces a larger release from the SR, that then is taken more efficiently out of the cell by NCX, resulting in a net efflux of calcium from the cell. This counterintuitive result (described experimentally in [26]) is not at all unique. It seems clear that understanding and predicting calcium homeostatic levels is key to understand the contractibility behavior of cardiac ventricles.

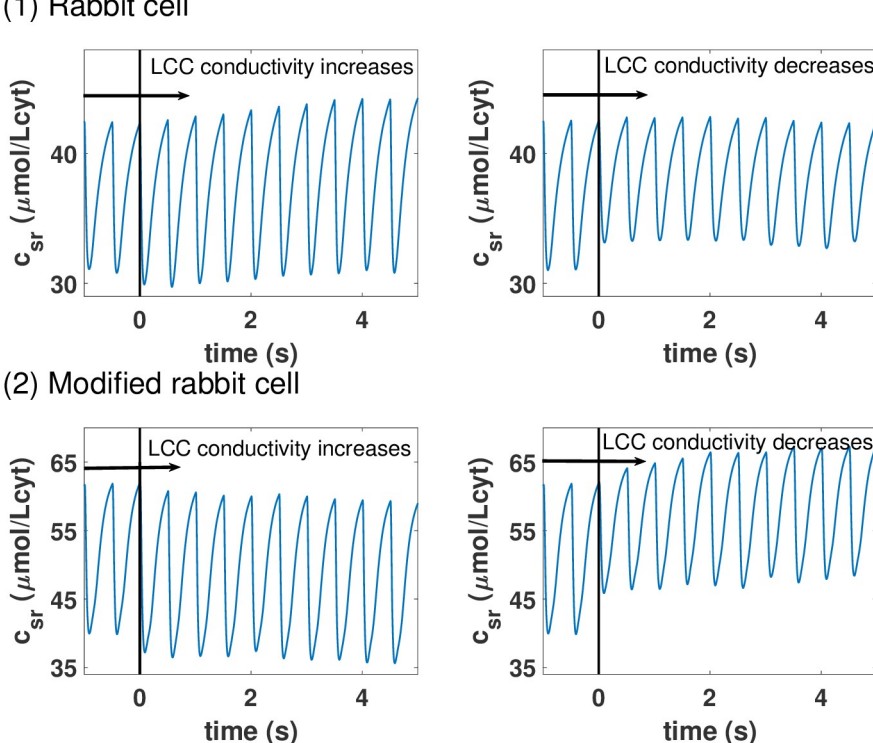

**Fig 2. Example of counterintuitive calcium homeostatic regulations.** Panel (1): Evolution of the SR free calcium concentration when, at a time marked by a black vertical line, the conductivity of the LCC is increased (left) or decreased (right). An increase in intake via the LCC leads to a cardiomyocyte with higher calcium average concentrations in the SR. Panel (2): Response of the cardiomyocyte to an increase in LCC conductivity, as in the first panel, when its buffering and SERCA uptake is different. Now, the increased intake via LCC is homeostatically regulated in a completely different way. The higher level of calcium intake leads to an even larger extrusion for the SR leading to a lower calcium load.

## Materials and methods

### Rabbit computational model structure

**General structure.**   We consider the cell to be a 3-dimensional array composed of $25 \times 25 \times 50$ Calcium Release Units (CaRUs) of volume $0.5 \times 0.5 \times 2\mu m^3$ following the basic structure described by Mahajan et al [27] (see first panel in Fig 1) where the cell is divided into volumes associated with RyR clusters. Ryanodine Receptors channels (RyR) are introduced following the description by Stern et al [28], as adapted in [29] with four possible states. We do not consider, for the L-type Calcium Channels (LCC), the states sensitive to Barium in [27]. We also introduce important modifications in the way we write the equations of the model to ensure calcium mass balance, so the difference between the calcium entering and leaving the cell matches exactly the change of calcium inside the cell. As in previous models [30], each CaRU is divided into 5 compartments or subvolumes, as shown in the second panel of Fig 1: cytosol (i), subsarcolemma (s), network sarcoplasmic reticulum (nsr), junctional sarcoplasmic reticulum (jsr) and dyadic junction (d), which is the space between a group of LCC channels, in the cellular membrane, and a cluster of RyR channels in the membrane of the sarcoplasmic reticulum. The basic structure of the CaRU is a rectangular cuboid with $dz$ being the distance between Z-planes, which we take to be roughly $2 \ \mu m$ [31, 32], while the distance in X-Y is the

average distance between large clusters of RyR (40) or the aggregation scales of smaller ones that we take to be 500 nm [33, 34].

**Rate equations in the CaRU.**   This model considers the diffusion of calcium ions between compartments of the same CaRU, as well as diffusion between neighboring CaRUs; moreover, there are other currents related to the dynamics of calcium buffers, thus differentiating between free and bound to buffers calcium ions in the different compartments. Finally, there are currents due to channels, exchangers or pumps. L-type calcium channels introduce calcium, $j_{LCC}$, in the dyadic space. Flux across RyR also goes into the dyadic from the junctional SR, $j_{RyR}$. SERCA pumps calcium $j_{SrCa}$ from the cytosol to the network SR and the Na$^+$/Ca$^{2+}$ exchanger is considered to be in the membrane and t-tubules, as LCC, but not close to it. We, therefore, consider that the exchanger flux, $j_{NCX}$, is sensitive to the calcium gradient between extracellular calcium and calcium concentrations in the subsarcolemma. The dynamics are deterministic except for the RyRs and LCCs which are defined by internal markovian states and whose transitions are considered to be stochastic.

The set of differential equations for Ca$^{2+}$ concentration in the different compartments on each CaRU (where we have omitted the $i, j, k$–th superscripts indexing the position of the CaRUs for simplicity) reads as follows, where we take no-flux boundary conditions:

$$\frac{dc_d}{dt} = j_{RyR} - j_{LCC} - j_{ds} \tag{1}$$

$$\frac{dc_s}{dt} = j_{NCX} - j_{si} + \frac{v_d}{v_s} j_{ds} + D_s \nabla^2 c_s \tag{2}$$

$$\frac{dc_i}{dt} = -j_{SrCa} + \frac{v_s}{v_i} j_{si} - j_{buff_i} + D_i \nabla^2 c_i \tag{3}$$

$$\frac{dc_{jsr_{TOT}}}{dt} = j_{tr} - \frac{v_d}{v_{jsr}} j_{RyR} \tag{4}$$

$$\frac{dc_{nsr}}{dt} = -j_{tr} + \frac{v_i}{v_{nsr}} j_{SrCa} + D_{sr} \nabla^2 c_{nsr} \tag{5}$$

where $j_{ds}, j_{si}, j_{tr}$ represent internal diffusion currents and $v_d, v_s, v_i, v_{jsr}$ and $v_{nsr}$ represent local volumes for dyadic space, subsarcolemma, cytosol, junctional SR and network SR respectively. We take $v_{jsr}$ to be equal to $v_{nsr}$ and reserve the notation $c_{sr}$ for the average concentration of $c_{nsr}$ and $c_{jsr}$, which can also be defined as the free SR calcium concentration. While in the model we track both, in our analysis, given that at the time scale of one beat both $c_{nsr}$ and $c_{jsr}$ equilibrate and become roughly one and the same with $c_{sr}$, we will often use the latter expressed in terms of liters of cytosol. We take the cytosolic volume to be half of the available space (0.25 $\mu m^3$) given the presence of mitochondria, which we neglect in this model.

$D_s, D_i$ and $D_{sr}$ are the inter-CaRU diffusion constants for subsarcolemma, cytosol, and SR respectively. In the case of subsarcolemmal diffusion, we consider that these volumes do not exist across z-planes so $(D_s)_z = 0$. Regarding $D_i$ and $D_{sr}$, they are different from each other, slower in the SR, but the same in all directions. Notice, however, that given that the characteristic length is different across the z-plane direction, the characteristic time scales of diffusion in the Z direction is different than in the X-Y direction.

The model spatial structure considers three different diffusive currents within a CaRU, which depend only on the difference of Ca$^{2+}$ concentration among the affected compartments

with a given characteristic time of diffusion. These are: diffusion between dyadic junction and subsarcolemma, $j_{ds} = (c_d - c_s)/\tau_d$; diffusion between subsarcolemma and cytosol, $j_{si} = (c_s - c_i)/\tau_s$; diffusion between network and junctional sarcoplasmic reticulum, $j_{tr} = (c_{nsr} - c_{jsr})/\tau_{tr}$.

**Pacing by external clamped potential: Calcium transient and recovery.** The above equations have to be complemented with an external clamped potential which fixes an external pacing. In this model, the pacing of the model is set by a periodic clamped action potential with a constant pacing period T and an action potential duration APD described in the Supporting information file S1 File. Each time voltage raises, the model reproduces the Calcium-Induced Calcium-release (CICR) mechanism at the global level from the stochastic behaviour of the RyR clusters, where the LCC opening raises the probability of RyR2 to open, releasing calcium from the SR. This results in an averaged behavior as the one depicted in the third panel of Fig 1. Average free calcium concentration values in the cytosol raise and drop during one beat (calcium transient), while in the SR a transient drop in the concentration of free calcium is followed by its recovery.

**Currents of rabbit model.** The four basic currents that allow for the normal calcium transient, namely intake of calcium by the LCC, extrusion by NCX, release by the RyR and uptake by SERCA, are fully detailed in the SM. Here we present their basic dependencies.

The current associated with the Na$^+$/Ca$^{2+}$ exchanger depends on intracellular and extracellular Na$^+$ concentrations ([Na]$_i$ and [Na]$_o$ respectively), which are set as constant in voltage-clamped model, as well as on extracellular Ca$^{2+}$ concentration ([Ca]$_o$), also constant, and subsarcolemmal calcium concentration $c_s$.

The current through the LCC channels, carrying calcium from the extracellular medium towards the intracellular space, depends on the transmembrane voltage, the extracellular Ca$^{2+}$ concentration and the calcium concentration in the dyadic volume close to both the RyR cluster and the LCC channels. The current is proportional to the number of LCC channels in the open state ($O_{LCC}$). We use the expression given in [27] for the properties of LCC in the rabbit, not taking into account those states related to barium inactivation. This leaves us with five possible states: O for the open state, C1 and C2 for closed states, and I1 and I2 for the inactivated states. In each CaRU, we consider a group of 5 LCC channels, as shown in the second panel of Fig 1. We track the state of each one of the channels using the length rule to establish a transition between states with the uniformly distributed random number generator RANMAR.

The calcium release current, through the RyRs, is diffusive, but depends not only on the difference among $c_{jsr}$ and $c_d$ but also on the conductivity of the RyR, which is set by the number of these proteins in the open state ($O_{RyR}$): $j_{RyR} = g_{rel} O_{RyR}(c_{jsr} - c_d)$. In each CaRU we consider a cluster of $N_{RyR} = 40$ RyR, where each one of the receptors can be in one of the 4 possible states shown in the second panel of Fig 1: O, for the open state; C is the closed state, while I1 and I2 are two inactivated/terminated states. Their dynamics are treated stochastically as a function of transition rates which depend, both for opening and inactivation, on calcium concentrations in both the dyadic and the jSR spaces (a detailed description of the transition rates in the RyR can be found in the Supporting information S1 File).

Finally, the current given by the SERCA pump is thermodynamically limited. It is considered to depend only on Ca$^{2+}$ concentrations in cytosol and network SR. We give the full expression here given the relevance for our analysis of SERCA gene therapy.

$$j_{SrCa} = v_{up} \frac{\left(c_i/K_i\right)^2 - \left(c_{nsr}/K_{sr}\right)^2}{1 + \left(c_i/K_i\right)^2 + \left(c_{nsr}/K_{sr}\right)^2} \tag{6}$$

where K$_i$ and K$_{sr}$ are half occupation binding constants, and $v_{up}$ is the maximum uptake strength whose effects on the model will be analyzed in detail.

**Calcium buffering implementation.**    Buffers are located by construction in the cytosol and in the junctional SR where calcium can bind to them. Therefore, the above equations have to be complemented with the equivalent dynamics equations for the calcium binding to the buffers. In this respect our model presents a very important difference with previous models. We do not consider the fast-buffering approximations as described in [35, 36] and used, for instance, in [37], since this leads to a loss in the mass balance in the dynamical equations and it does not particularly speed up the code. In our analysis, it is critical to have an algorithm that strictly balances mass to all orders. This leads us to consider jSR to be the volume around the RyR where calsequestrin is present and to compute the evolution of $c_{jsr_{TOT}}$, which includes both free calcium in jSR ($c_{jsr}$) and calcium bound to calsequestrin in that compartment. The reason is that calsequestrin is the only buffer that is fast enough to slow down the performance of the code if the binding of calcium to the buffer is computed dynamically. Once the total amount of calcium is computed in the jSR, this calcium is split between the free and buffered parts using the fast-buffering approximation

$$c_{jsr_{TOT}} = c_{jsr} + \frac{c_{jsr}B_{CSQ}}{c_{jsr} + K_{CSQ}} \tag{7}$$

where $B_{CSQ}$ is the concentration of calsequestrin in the jsr and $K_{CSQ}$ its dissociation constant. The free calcium concentration $c_{jsr}$ can be obtained solving the quadratic equation:

$$c_{jsr} = \frac{1}{2}\left(c_{jsr_{TOT}} - (K_{CSQ} + B_{CSQ}) + \sqrt{(K_{CSQ} + B_{CSQ} - c_{jsr_{TOT}})^2 + 4K_{CSQ}c_{jsr_{TOT}}}\right) \tag{8}$$

All the other buffers have their own dynamical equation where the calcium taken out from its free level goes to the calcium bound to the buffer. Time scales and affinities are mostly taken from Shannon et al. [38]. A full description is incorporated in the Supporting information S1 File.

**Parameters of the rabbit model and its modified version.**    A full list of parameters of the rabbit model including buffers is provided in the appendix of the Supporting information S1 File. Furthermore, this appendix also includes the changes in the parameters of the modified rabbit model used in Fig 2. This modification of the rabbit model allows us to show in the introduction that the effects of increasing the conductivity of LCC can depend strongly on the animal model. We must stress that we change the parameters of our rabbit model but not its structure. More specifically, our modified rabbit model has different buffer levels in order to change the effect on the level reached by calcium in the transient and a slightly reduced maximum SERCA uptake to mimic a slight reduction in its expression.

## General equilibrium method to study calcium homeostasis

We develop here a novel method to study calcium homeostasis. We proceed to show that a bi-dimensional general equilibrium approach provides the proper framework to understand and predict intracellular calcium levels at steady state. The actual homeostatic calcium level observed in an in-silico rabbit ventricular model can be predicted using a dramatic reduction of dimensionality, where the general equilibrium of thousands of variables is effectively reduced into a general equilibrium of two variables, namely, the level of free calcium in the cytosol and the level of free calcium in the SR. The reduction of the dimensionality allows us to show that calcium homeostasis can be understood in terms of shocks as in basic macroeconomic models and unveil that changes in SERCA function are complex double shocks. This opens the possibility of understanding calcium homeostasis from this new perspective.

As described in the previous section, we consider a subcellular compartmental model of calcium handling, dividing the cell into CaRUs, each incorporating a cluster of RyRs and of LCC, whose gating follows a Markov chain, solved stochastically. Let us denote the current of calcium that goes into the cell through the LCC channels associated with the $i$th CaRU as $j^i_{LCC}$, where now this current is defined by liter of cytosol ($\mu$mol/(Lcyt ms)). The average flux of calcium entering the cell via LCC $J_{LCC}$ ($\mu$mol/(Lcyt ms)) is:

$$J_{LCC} = \frac{1}{N}\sum_i j^i_{LCC} \tag{9}$$

where $N$ is the number of CaRUs in the cell. Integrating during the period $T$ we obtain the total amount of calcium that enters/intrudes the cell during one beat:

$$\Delta Q_{in} = \int_0^T J_{LCC}\, dt. \tag{10}$$

We can define similarly the average flux of calcium that leaves/extrudes the cell via the Na-Ca exchanger $J_{NCX}$ and the total amount of calcium that leaves the cell $\Delta Q_{out}$.

$$J_{NCX} = \frac{1}{N}\sum_i j^i_{NCX}, \quad \Delta Q_{out} = \int_0^T J_{NCX}\, dt. \tag{11}$$

Notice that both $\Delta Q_{in}$ and $\Delta Q_{out}$ depend, in principle, on the value of the roughly one hundred thousand internal variables $\phi^i_j$ (where $j$ stands for the number of open LCCs, amount of Ca bound to buffers, etc, at each of the $i$ different CaRUs) that define the state of the cell at the beginning of the beat. We assume that, despite the stochastic nature of LCC and RyR2 channels, average values are well-behaved. This being the case, we can establish that the total amount of calcium $Q^n_T$ at beat $n$ follows the relation $Q^{n+1}_T = Q^n_T + \Delta Q^n_{in} - \Delta Q^n_{out}$ so that at steady state (*ss*) we have:

$$\Delta Q^{ss}_{in}(\phi^i_j) = \Delta Q^{ss}_{out}(\phi^i_j) \tag{12}$$

The total amount of calcium released via RyR2 from the SR to the cytosol, $\Delta Q_{rel}$, and the total calcium uptaken by SERCA, $\Delta Q_{up}$, can be equally defined from the average fluxes of all individual RyR clusters, $J_{RyR}$, and SERCA pumps, $J_{SrCa}$:

$$\Delta Q_{rel} = \int_0^T J_{RyR}\, dt \qquad \Delta Q_{up} = \int_0^T J_{SrCa}\, dt \tag{13}$$

This gives a beat-to beat change in the SR calcium concentration given by:
$Q^{n+1}_{SR} = Q^n_{SR} + \Delta Q^n_{up} - \Delta Q^n_{rel}$. In steady state, the amount of calcium in the SR must be constant, leading to

$$\Delta Q^{ss}_{rel}(\phi^i_j) = \Delta Q^{ss}_{up}(\phi^i_j) \tag{14}$$

As stated before, ventricular myocytes reproduce consistent average behavior. Therefore, fluxes in Eqs (12) and (14) depend on the relevant spatial average variables $\bar{\phi}_j$ and not on their spatial distribution.

Except for average calcium concentrations, all the average quantities that define the state of the cell under voltage-clamped conditions have relaxation time scales which are shorter than the fastest pacing period reached. This typically goes from $T_{min} = 300 - 400$ ms in humans to $T_{min} = 150 - 200$ ms in rats. With this in mind, we notice that LCC, exchanger, SERCA and

buffer characteristic time scales are typically faster than $T_{min}$. Recovery times of the RyR2 are probably around 100 ms, safely below $T_{min}$ for most species.

Thus, we claim that it is possible to do a separation of time scales, such that the dynamics of the homeostatic process occurs in a slow manifold, where the dynamics of the fast variables slaves to that of the slow variables (or order parameters, in synergetic terminology [39]). We, thus, divide the variables into fast, $\phi_{fast}$, and slow, $\phi_{slow}$. The latter equilibrate in a time scale of several beats. At each individual beat, however, the former attain their equilibrium values, that will depend on the state of the slow variables, so $\phi_{fast}^n = h(\phi_{slow}^n)$. This leads us to the conclusion that the surface which determines homeostasis suffers a huge effective reduction of its dimensionality since memory is mainly eliminated by the external pacing. Flows in one beat are then determined only by the diastolic free SR concentration $c_{sr}$ and cytosolic concentrations $c_i$, that act as slow variables, such that one can compute effective maps for the total amount of calcium in the cell and in the SR, as:

$$Q_T^{n+1} = Q_T^n + \Delta Q_{in}^n(c_{sr}, c_i) - \Delta Q_{out}^n(c_{sr}, c_i) \tag{15}$$

$$Q_{SR}^{n+1} = Q_{SR}^n + \Delta Q_{up}^n(c_{sr}, c_i) - \Delta Q_{rel}^n(c_{sr}, c_i) \tag{16}$$

Maps as the ones indicated in Eq (16) for the concentration of calcium in the SR have been already constructed and used, for instance, to study the stability of the dynamics to alternans [37, 40, 41]. However, in previous work, this map does not include the dependence on cytosolic calcium concentrations since they do not consider homeostatic equilibrium. The models have one global variable and one map to equilibrate. In order to treat the homeostatic problem, we need to supplement it with the map for the global calcium level in Eq (15) and clarify the double dependence on cytosolic and SR calcium levels.

Then, at steady state, we have:

$$\Delta Q_{in}(c_{sr}^{ss}, c_i^{ss}) = \Delta Q_{out}(c_{sr}^{ss}, c_i^{ss}) \tag{17}$$

$$\Delta Q_{rel}(c_{sr}^{ss}, c_i^{ss}) = \Delta Q_{up}(c_{sr}^{ss}, c_i^{ss}) \tag{18}$$

These are two equilibrium conditions for two independent global average variables. If a steady state exists, it corresponds to the simultaneous fulfillment of the two global equilibrium conditions. Alternatively, the solution to these equations may either not exist or, if it does, be unstable, corresponding to, for instance, alternans or chaotic behavior. In this paper, we address the first scenario with a stable fixed point.

We provide further information regarding these maps in the Supporting information S1 File where we show that the total calcium concentrations in the cell and the SR could also be treated as the fundamental analytical variables instead of free calcium concentrations. The reason is that end-diastolic free calcium concentrations can be related to total calcium concentrations assuming the equilibrium concentration of calcium bound to buffers. So, even though in the mathematical model we only use the fast buffering approximation for calsequestrin, the other buffers equilibrate so fast that they attain the equilibrium values by the end of each period.

## Results

### Validation

From Eqs (17) and (18) we can predict the average diastolic calcium level at steady-state from measurements away from steady-state. Figs 3 and 4 sketch the basic procedure to predict the

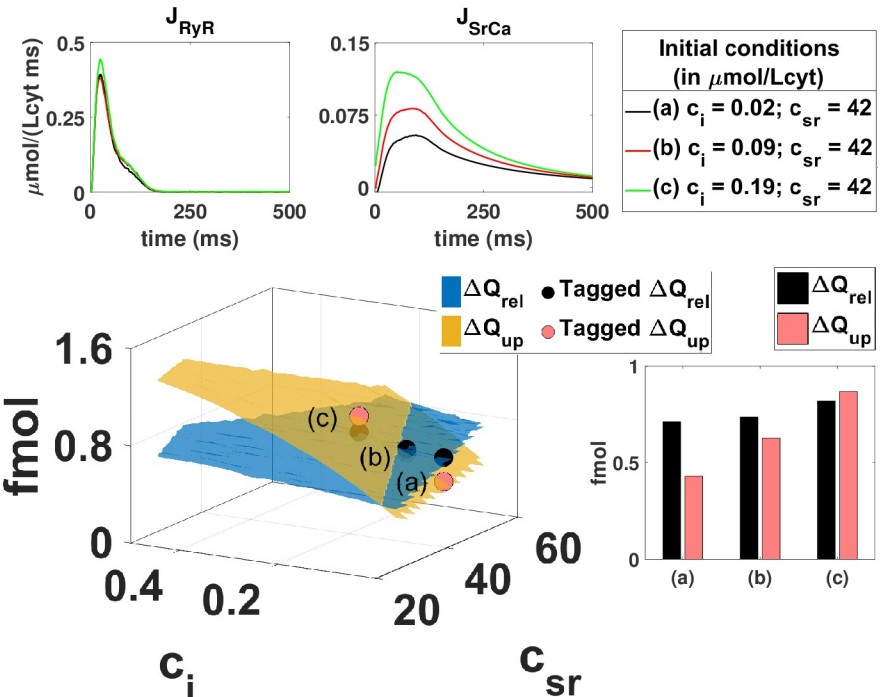

**Fig 3. Schematics of the procedure to reconstruct one of the nullclines of the system.** The nullcline corresponds to the partial equilibrium where the intake of calcium into the SR equals its release, where we have multiplied all $\Delta Q$ by the volume of the cytosol, to obtain this value in femtomols (fmol). The main central graph reproduces total release $\Delta Q_{rel}$ and uptake $\Delta Q_{up}$ as a function of the initial free concentrations in the cytosol $c_i(t=0)$ and SR $c_{SR}(t=0)$. The nullcline is determined by the line where both surfaces cross. For each initial condition, one single transient is simulated and the release $J_{RyR}$ and uptake $J_{SrCa}$ computed and shown in the upper graphs. The integrated values are indicated in the bar graph below and then placed as elements of the surface. The whole surface is constructed by reproducing this procedure with multiple different initial calcium concentrations where all other initial variables are in their fast equilibrium approximation.

steady state from single beat measurements. We simulate the evolution of calcium concentrations, $c_i$ and $c_{sr}$, during one single beat, taking different initial calcium diastolic values (i.e. different initial conditions), with all the local variables unrelated to calcium concentrations set initially at their fast-equilibrium approximation. From this evolution we can compute how much calcium leaves the SR, $\Delta Q_{rel}$, or enters the SR, $\Delta Q_{up}$, during this beat, and repeat the calculation for a different initial condition. This gives new values of $\Delta Q_{rel}$ and $\Delta Q_{up}$. If we repeat this procedure for multiple combinations of initial values of $c_i$ and $c_{sr}$ we can reconstruct the dependence of the currents with the diastolic calcium concentrations. We obtain the functions $\Delta Q_{rel}(c_{sr}, c_i)$ and $\Delta Q_{up}(c_{sr}, c_i)$, which are nothing else than surfaces in a diastolic calcium concentration space as shown in Fig 3.

It is rather intuitive that release and uptake depend on cytosolic and SR calcium concentration at the beginning of one beat. Less intuitive is that intake and extrusion also depends on $c_{sr}$ even though the L-type Calcium current and exchanger expressions do not depend instantaneously on it. The reason for this strong dependence on the SR calcium concentration at the beginning of the beat is because a larger or smaller initial calcium SR load leads to larger or smaller cytosolic calcium transients during the beat. A different calcium transient results, in general, in a different inactivation in the LCC and NCX extrusion during that beat.

The line where both surfaces cross (see $\Delta Q_{rel} = \Delta Q_{up}$ in Fig 3), or g-nullcline, fulfills the partial equilibrium indicated by Eq (17). To obtain the partial equilibrium given by Eq (18) we

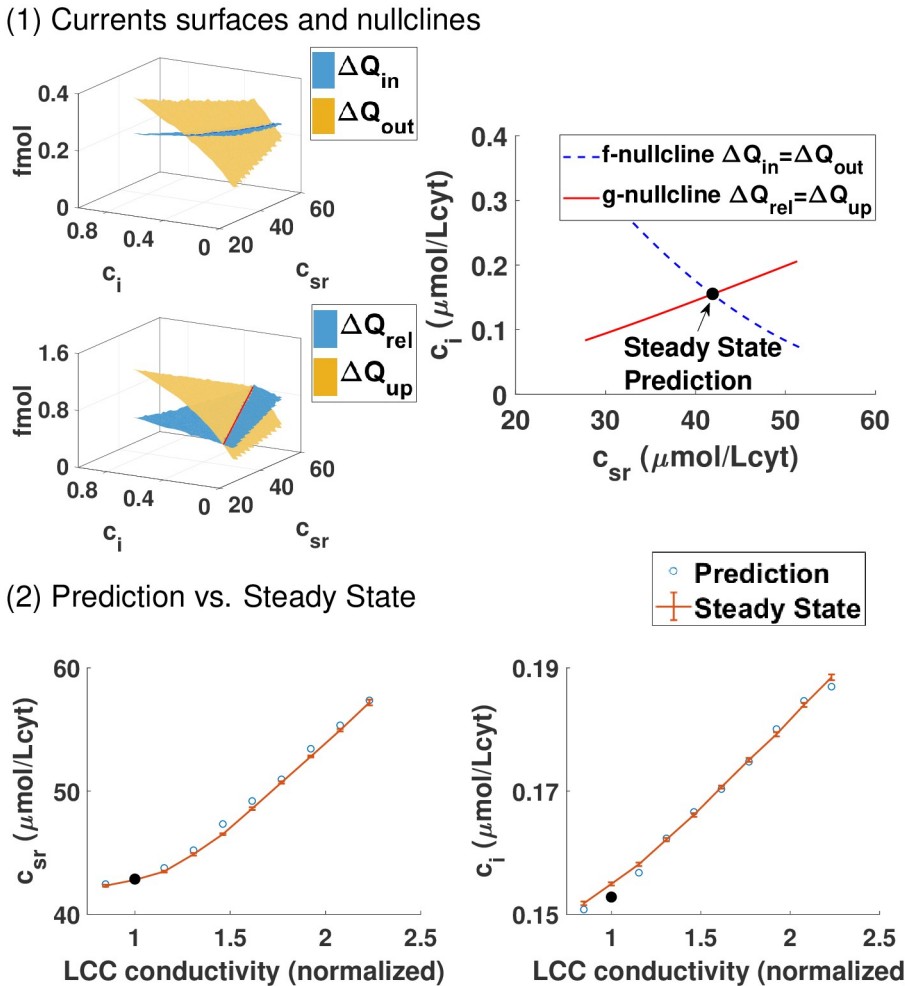

**Fig 4. General equilibrium. Prediction of steady-state from the nullclines crossing.** Panel (1): The first two surfaces indicate the total amount of calcium entering $\Delta Q_{in}$ and leaving $\Delta Q_{out}$ the cell, computed following the same procedure explained in Fig 3. Below we compute the surfaces corresponding to release and uptake of calcium from the SR. On the right, we plot the two nullclines that correspond to the crossing of the previous surfaces. The f-nullcline is the set of initial $c_i$ and $c_{sr}$ where $\Delta Q_{in} = \Delta Q_{out}$ while the g-nullcline corresponds to the crossing between release $\Delta Q_{rel}$ and uptake $\Delta Q_{up}$. The point where both nullclines cross gives the steady state of the system $(c_i, c_{SR})$ where concentrations return to the same pre-systolic values after a stimulation. Notice that the parameters of the model fix this crossing precisely at the expected value of roughly 150nM in the cytosol and 40 $\mu$mol/Lcyt in the SR, which correspond to a local concentration of free calcium in the SR $c_{sr}$ at roughly 500 $\mu$M, given the volume ratio between SR and cytosol. We notice that this free calcium concentration corresponds to 100 $\mu$mol/Lcyt of total calcium in the SR, free and bound to CSQ, as reported in [42]. Panel (2): Steady state from single beat measurements at 2 Hz described in the top panel with modification of the rabbit model where the conductivity of LCC is increased. The prediction of thesteady state obtained from the global equilibrium as a function of the LCC conductivity fits very well the computed steady-state obtained after letting the system evolve for 100 beats.

construct the surfaces for $\Delta Q_{in}(c_{sr}, c_i)$ and $\Delta Q_{out}(c_{sr}, c_i)$ using the same procedure indicated in Fig 3. The line where they cross is the f-nullcline, which represents the equilibrium condition that at steady state the same amount of calcium that enters the cell must leave it. The point where both nullclines intersect gives us the concentration of diastolic calcium that fulfills both equilibrium conditions. This sets the steady state, i.e., it predicts the diastolic homeostatic calcium level and its internal distribution between the SR and cytosol of the cell.

We have checked that the prediction given by the crossing of the nullclines agrees with the calcium concentration values obtained letting the system evolve to steady-state (bottom panel of Fig 4). This agreement confirms the validity of the reduction in dimensionality. This reduction is not at all trivial. It comes from the fact that average values are not very sensitive to the inherent noise of the system in ventricle myocytes and that the processes involved in calcium handling have time scales that are faster than the pacing period. Regarding calcium homeostasis, each period of the system provides a global reset of the fast variables leaving only the slow concentration variables in play.

## General shocks

Our key result has deep implications for the analysis of cardiomyocyte contractibility. We can use this framework to infer how the cell behavior will be modified upon a change in channel properties or the increase of buffers levels. More specficially, we can use the insights from bidimensional general equilibrium problems from other fields and understand the counterintuitive response of cells to changes at the channel level, as described in the introduction. We will focus our discussion on changes in SERCA function given its possible relevance to understand gene therapy failure. We will show that, in terms of shocks, as they are normally called in the literature, a change of SERCA strength is actually a double shock. With this realization in mind we proceed to unveil and discuss a physiological mechanism for SERCA gene therapy failure where, even in the case of SERCA improved function, calcium transient and contractibility are not improved.

Analysis of stability based on partial equilibrium conditions is common in other fields, such as macroeconomics. In fact, this framework can be mapped one-to-one to the Investment-Saving, Liquidity preference-Money supply (IS-LM) model of macroeconomics developed by J. Hicks [2] as an explanation of Keynes general theory of Employment, Interest and Money [43]. In the IS-LM, the steady-state of a large complex closed economy is defined by its ouput (average GDP per capita) and by the average interest rate, just like in our case we define our steady state by two variables, the average calcium concentration in the cytosol and the SR. Equally, the general equilibrium conditions that must be met are two. First, Investment has to balance Saving, leading to the IS curve, equal to our f-nullcline. The equivalent of our g-nullcline involves the interest rates and output for which the money market is in equilibrium.

There, a shock is a sudden change in the curves that determine equilibrium. In our case, a change in the conductivity or the property of a particular receptor or the strength of SERCA/NCX is a shock that changes the four surfaces corresponding to the four fluxes. If the shock were only to affect strongly the surfaces related to the release and uptake of calcium from the SR (as sketched in the first panel of Fig 5), then just the g-nullcline would shift position. From this shift and the slope of the f-nullcline we can predict the effect of the shock. For example, if the uptake of calcium is raised (increasing the $\Delta Q_{up}$ surface) then the g-nullcline will shift down. Given that, in this example, the slope of the f-nullcline is negative, the downward shift of the g-nullcline will move the crossing of the two nullclines, and the steady state, to higher $c_{sr}$ and lower $c_i$.

However, as we proceed to describe in the next sections, changes in key functioning elements of calcium homeostasis normally affect two or all four of the surfaces. To be specific, we are going to consider a change in SERCA function.

## Consequences of shocks

We aim to understand how an increase in SERCA function affects the homeostatic state and whether it systematically leads to improved calcium transients and contraction. We will model

## (1) Mechanics of a single shock

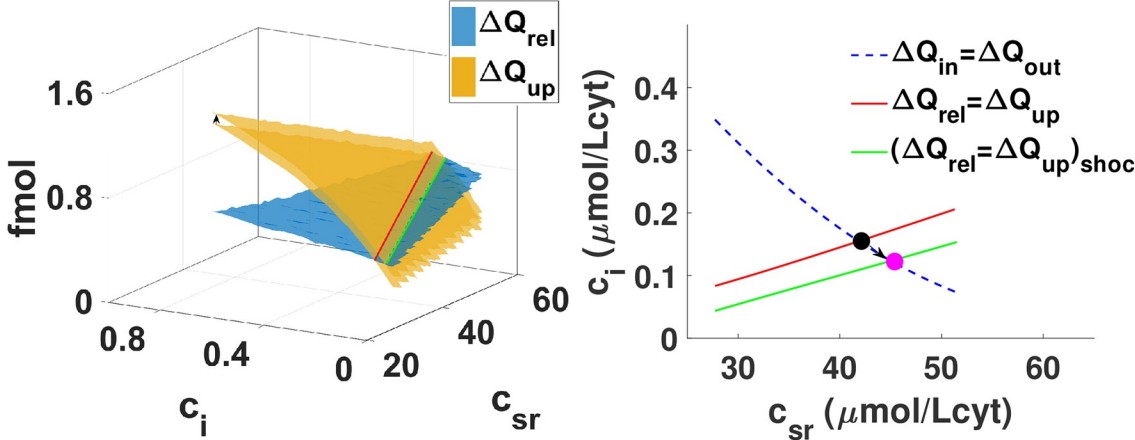

## (2) Increasing SERCA uptake actually leads to a double shock

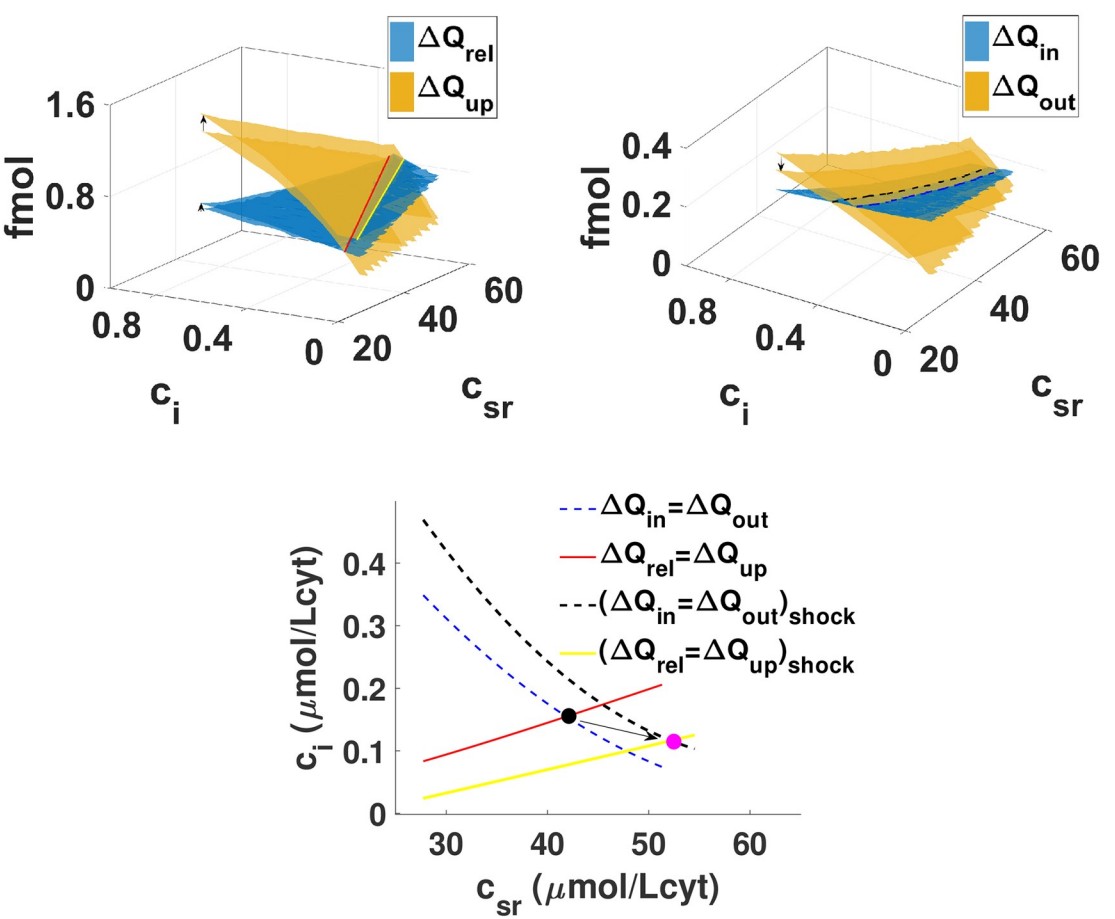

**Fig 5. Shocks in general equilibrium models. Change in SERCA uptake is a two-nullcline shock.** Panel (1): We show a hypothetical change in the cell that only affects one of the four surfaces. This produces a change in one of the nullclines. As an example, we pick an increase of the $\Delta Q_{up}$ surface. This leads to a shift of the g-nullcline to the right compared with the previous case, resulting in a lower diastolic level in the cytosol but larger in the SR. Panel (2): Effect of changing the maximum uptake of the SERCA pump from 0.15 $\mu$M/ms to 0.3 $\mu$M/ms in our rabbit model. Increasing the function of SERCA does not only affect the g-nullcline, expected since the uptake is larger, but also affects how well the exchanger works shifting also the f-nullcline to the right. The end result is similar to the previous case.

this increase in SERCA function with an increase in its maximum uptake rate $v_{up}$. This change mimics increases/decreases in the number of pumps present in each of the CaRUs. This is related to the number of pumps expressed in the SR membrane and should increase if SERCA therapy works properly. We proceed to answer whether a change of SERCA function always leads to larger/broader calcium transients.

**Change in SERCA function is a double shock.**    We observe that changes in $v_{up}$ affect not only the surface directly related to the uptake $\Delta Q_{up}$, but also $\Delta Q_{out}$. Thus, a change in SERCA function is a double shock, since it changes two nullclines, as shown in the bottom panel of Fig 5. The reason is rather intuitive. A change in the uptake does not have a strong effect in the behavior of the LCC channels and hardly changes the release, but a more efficient SERCA reduces quickly cytosolic calcium levels, which reduces the efficiency of the exchanger. The effect is that the shock associated with larger uptake moves both nullclines to the right. Given the structure of the nullclines, shown in Fig 5, where the f-nullcline has a negative slope while that of the g-nullcline is positive, it is thus straightforward that they will cross at a higher level of SR. In fact, we observe that the level of calcium in the SR increases monotonically, while the level in the cytosol decreases also monotonically (left of the first panel of Fig 6). In the Supporting information S1 File, we show that the total amount of calcium in the cell, this is, the sum of calcium in the cytosol and SR, increases slightly as we increase the SERCA uptake given that the increase in calcium in the SR compensates a reduction of calcium levels in the cytosol. While the increase in SR load depends on the slope of the nullclines, whether pre-systolic calcium in the cytosol increases or decreases depends on how far both nullclines move with the shock. Thus, the fact that the nullclines cross at lower cytosolic calcium is not a structural result but it depends on the particulars of the system.

The shift of the nullclines due to the increase in SERCA function is a quantification of how SERCA interacts with the exchanger and, indirectly, via changes in the release, with the LCC. A faster SERCA uptake reduces the ability of the exchanger to extrude calcium since SERCA pushes down calcium in the cytosol faster during diastole, making the gradient of calcium between the cytosol and extracellular space larger (see Fig 7). Given that the exchanger does not work as well with a large gradient in calcium, one could think that it will extrude less calcium. However, this cannot be the case. Since $\Delta Q_{in}$ is a rather flat surface, calcium entering via LCC, as SERCA increases, remains roughly unchanged. Thus, to reach equilibrium, calcium extruded by the exchanger has to remain almost constant too. The only possibility is that the system reaches a different homeostatic equilibrium in which the release and calcium transient adapt so that the exchanger can extrude the same as it did before the shock (Fig 7).

The resulting changes in the calcium transient when we increase the maximum SERCA uptake at steady-state are summarized on the right graph of the first panel in Fig 6 and explained schematically in Fig 7. The transient amplitude must change if diastolic cytosolic calcium decreases. We observe that the peak of calcium transient barely changes, but given that the minimum of the transient decreases, the amplitude raises appreciably. The end result is that in this animal model, a higher $v_{up}$ increases the contractibility of the cell, given the larger transient.

**Physiological mechanism for SERCA gene therapy failure.**    Now we aim to answer a simple question. Will a change of SERCA function always lead to larger/broader calcium transients? For that, we will reproduce a possible malfunction. Given that there are calcium-binding proteins that affect both SERCA and RyR [19], we consider a rabbit cell where the conductivity of the RyR is reduced without changes in the exchanger or the LCC. The details of the rabbit cell with reduced RyR activity are explained in S1 File where we show that the slopes of both nullclines are still the same as in the original rabbit model. However, as the second panel of Fig 6 shows, there are crucial differences with the wild-type scenario, mainly that

## (1) Changing SERCA uptake in rabbit model

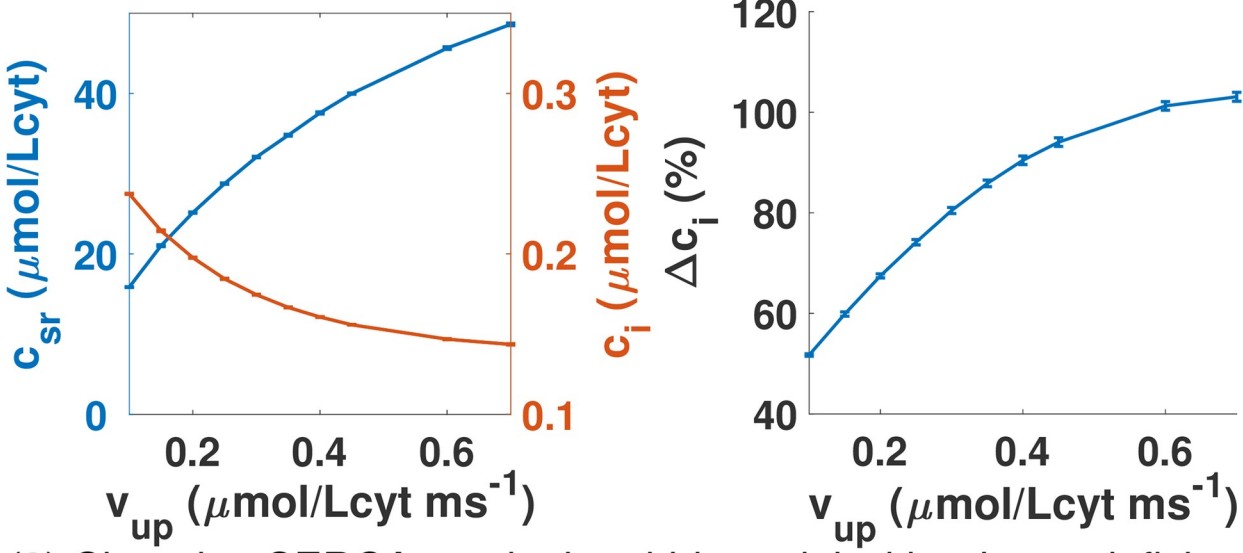

## (2) Changing SERCA uptake in rabbit model with release deficiency

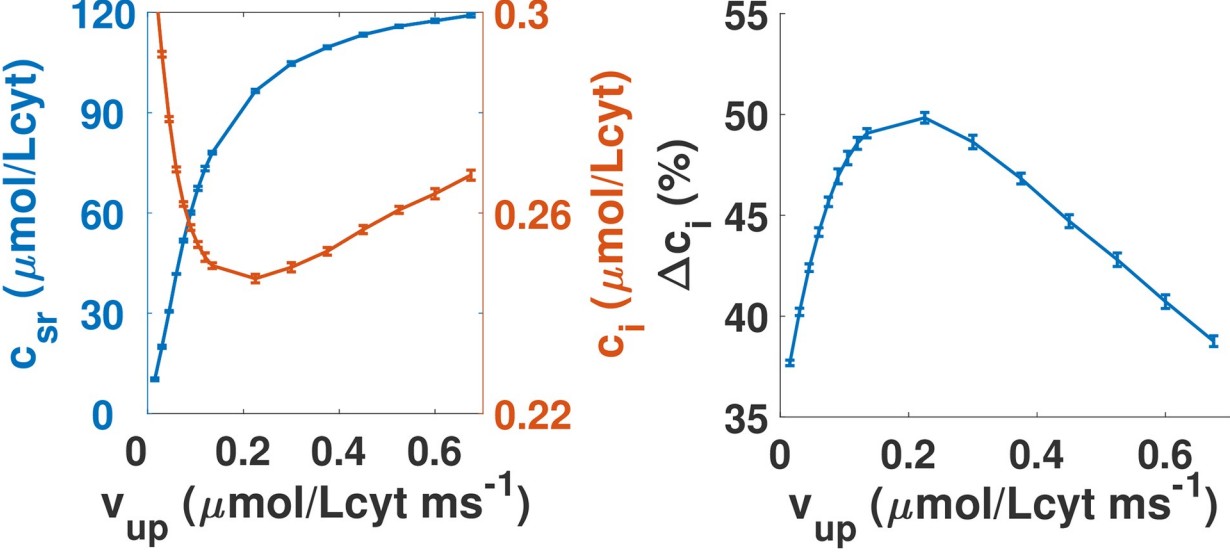

**Fig 6. Homeostatic response of increasing SERCA uptake with normal and low RyR channel conductivity.** Panel (1): Dependence of pre-systolic values of free calcium concentrations in the SR and in the cytosol at steady state with maximum uptake of SERCA pump $v_{up}$ in the rabbit model. On the right, the relative increase of calcium during the calcium transient as a function of the maximum SERCA uptake $v_{up}$. We take 100% to be the transient with the standard $v_{up} = 0.5\ \mu M/ms$. Panel (2): We present the same graphs as in the first panel but for a cell where the single channel conductivity of the RyR has been reduced. In this situation, the calcium transient is always lower than before and it does not improve as the uptake of SERCA is larger.

the calcium transient is actually reduced leading to lower contraction when SERCA maximum uptake is increased.

Given that the structure of the nullcline is the same, increasing the SERCA uptake in the RyR2 modified rabbit cell leads to higher diastolic SR load (see Fig 6). However, as shown in the left graph of the second panel in Fig 6, higher $v_{up}$ leads to an increase of the diastolic

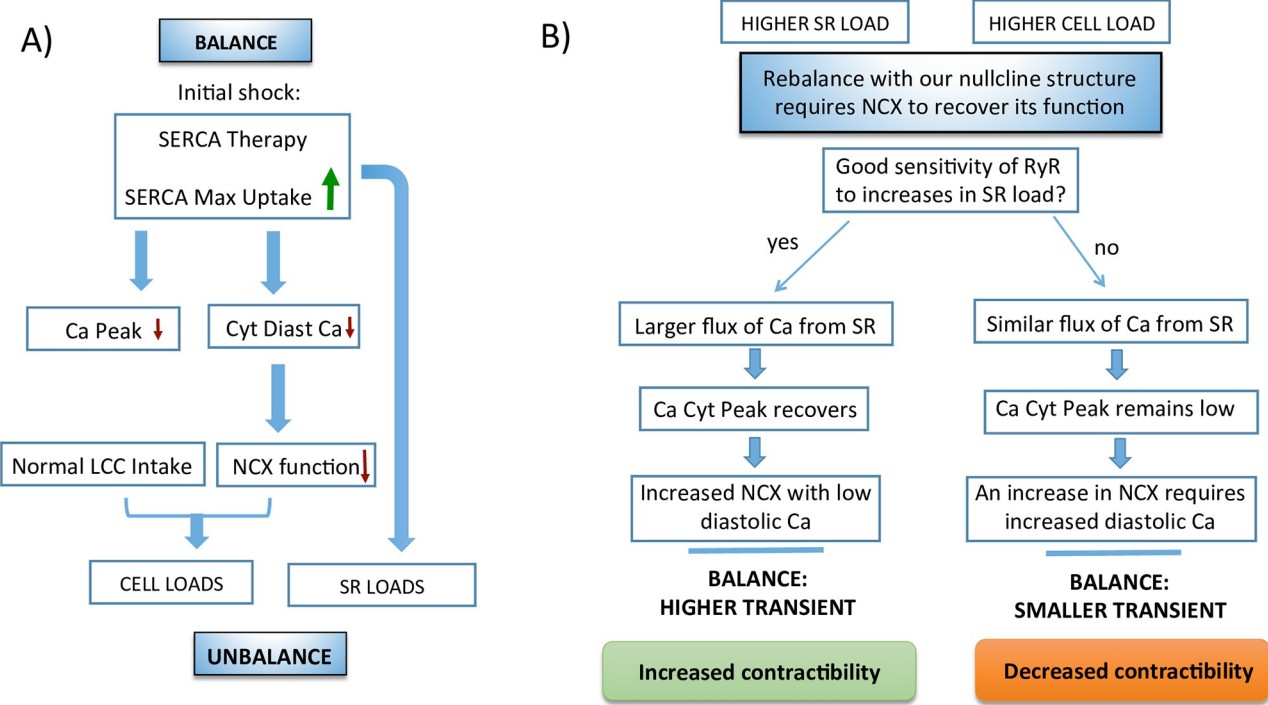

**Fig 7. Schematics of possible SERCA gene therapy outcomes.** Schematics of how SERCA gene therapy may fail in certain cells. A) During SERCA gene therapy a sudden increase in maximum SERCA uptake does not change initially the LCC or the release but it decreases NCX function as it decreases cytosolic calcium. This necessarily leads to a state of calcium unbalance that must be rebalanced again. B) Homeostasis will always fix and rebalance the system. However, the particulars of how this is done are not universal. They depend on the nullcline structure and reaction to shocks. For instance, in a healthy rabbit, the cell increases total calcium content by increasing considerably the calcium in the SR while diminishing cytosolic calcium levels. This increase in SR calcium results in a higher transient that allows a higher extraction of calcium through the NCX, even if diastolic calcium is reduced. However, it can happen that the RyR has a weak sensitivity to SR calcium content, so release remains almost constant. Since uptake is increased, the diastolic cytosolic level has to be increased to allow calcium extrusion through the NCX, then the transient is decreased, which results in decreased contractibility despite a stronger SERCA.

cytosolic calcium. The reason is that, in this case, the shift in the f-nullcline is larger than in the g-nullcline (see S1 File for details). This increased level of calcium makes the exchanger work more efficiently. On the other hand, the surface $\Delta Q_{in}$ is very flat, so the calcium influx remains almost constant. This means that the exchanger has to extrude roughly the same amount of calcium no matter what the SERCA uptake is. This directly leads to a decrease of the transient, as shown at the graph on the right of the second panel in Fig 6 and explained schematically in Fig 7. Actually, both the minimum is increased and the maximum is reduced at steady-state, leading to a strongly reduced transient, exactly the opposite of the wild-type case.

The sketch of the physiological mechanism for SERCA gene therapy failure is described in detail in Fig 7. Upon an increase in SERCA uptake, initially calcium intake via LCC and release via RyR remain unaffected. However, the NCX will be affected by a larger gradient in calcium leading to a reduction in its function. The system is no longer in equilibrium and must rebalance loading calcium (Fig 7A). This change in calcium levels affects all elements of calcium handling, including those that were not initially affected: intake and release. The specifics of how this increase in calcium is distributed depends on the particulars of the general equilibrium state of the cell. The slope of the nullclines will determine if SR will be more loaded as it is normally the case. The effect of the shock on the nullclines will determine whether cytosolic calcium is increased, because the cell will load until NCX function is restored thanks to a reduced gradient, or decreased, in case the NCX function is restored thanks to a larger

transient provided by a more loaded SR (Fig 7B). Both are perfectly possible outcomes of the shifts in the nullclines given the shock given to the cell.

## Discussion and conclusion

Detailed models of calcium handling have been able to reproduce spatiotemporally complex behavior such as discordant alternans [44] or calcium waves [45, 46], and have been used, for instance, to understand the onset of alternans as a synchronization transition [30]. Surprisingly given the complexity of the dynamics, the equilibrium conditions can be obtained using, as in very simple models, average variables and balance of fluxes in the cytosol and the SR. This result is not trivial. It requires that all internal variables, except cytosolic and SR calcium, equilibrate at the time scale of the pacing period. Also, that all CaRUs behave on average as the averaged variable, to prevent the appearance of intracellular inhomogeneities with different behavior.

As we have discussed earlier in this paper, this dimensionality reduction is not uncommon in other complex systems, and has been used previously to study the appearance of alternans in calcium cycling [37, 40, 41]. We understand the equilibrium point as satisfying two partial equilibrium conditions: calcium in and out of the cell and in and out of the SR must balance. We can then use this description to understand the effects of shocks, i.e., changes in parameters that result in shifts in the curves denoting the partial equilibrium conditions. We observe that the effect of the shocks can be very different depending on the form of these curves. This is, the same recipe may have very different outcomes depending on the state of the system.

In this paper, we have considered a clamped AP because the APD generally adapts to changes in currents and clamping does not affect this main insight. If we were to take the proper shape of the APD for a given frequency as our clamped AP in the model, the nullclines depicted here would not be affected. However, for future work, it would be interesting to study the interplay of calcium homeostasis with changes in the AP at different frequencies in order to address other cardiac properties such as the structure of force-frequency relations in different animal models. Since most of the adaptation of the AP is fast, we expect it to be generally slaved to the dynamics of calcium, at least, at the time scale of seconds. In this respect, a full linear stability analysis of the periodic transient, with an explicit calculation of the most unstable (or less stable) eigenvalues and eigenmodes [47, 48] would help to validate this point. However, there are slower times scales, associated with the long term accumulation of ions, that will become another dynamical variable, increasing the complexity of the general equilibrium problem. More specifically, one should expect the slow change of potassium and sodium concentrations to affect the LCC and NCX, which in turn will affect the nullclines which will feedback into the APD and back again into the ionic balance. The general structure of our approach will hold, but a full analysis of the relevant slow variables in the full model will be needed.

The analysis of a ventricular model in terms of shocks is thus robust, and we expect it to be useful to understand how these shocks may produce counterintuitive results. As, for instance, a decrease in SR calcium concentration as the strength of the LCC channels is increased. This can be well understood with the shifts in the partial equilibrium curves. Clearly, a further investigation of what determines the slope of the curves and how they depend on species or underlying pathologies should be undertaken.

Another relevant example is the effect of a change in SERCA. Upregulation of SERCA gene expression, or SERCA gene therapy, has been used to treat patients with heart failure. Even if this therapy is often successful, in some occasions it does not seem to work [22, 23]. From our results, one could think of a possible explanation as sketched in Fig 7. Under some

modifications on the parameters, an increase of SERCA strength does not lead to an increase in the cytosolic calcium transient, but rather the opposite. This is an in-silico failure of SERCA therapy due to how the cell reacts to the double shock that the changes in SERCA represent. We have thus unveiled a possible physiological mechanism for SERCA therapy failure and how its success or failure depends on the basic structure and movement of the nullclines determining homeostatic steady-state upon changes in SERCA function.

## Supporting information

**S1 File. Supporting information file with appendix.** In this supplemental information file, we give a full description of the rabbit model, develop the nullcline analysis, mentioned in the results section, using total calcium concentrations in the cell instead of free calcium concentrations. We also provide additional details on how SERCA uptake changes affect cell behavior concerning the physiological mechanism for SERCA gene therapy failure, as indicated in the discussion section. This supplemental file also includes an appendix with tables of the different parameters used in the models.
(PDF)

## Acknowledgments

We want to thank L.Hove-Madsen and N.Otani for fruitful discussions.

## Author Contributions

**Conceptualization:** David Conesa, Blas Echebarria, Yohannes Shiferaw, Enrique Alvarez-Lacalle.

**Formal analysis:** David Conesa, Blas Echebarria, Yohannes Shiferaw, Enrique Alvarez-Lacalle.

**Funding acquisition:** Blas Echebarria.

**Investigation:** David Conesa, Blas Echebarria, Angelina Peñaranda, Inmaculada R. Cantalapiedra, Yohannes Shiferaw, Enrique Alvarez-Lacalle.

**Methodology:** David Conesa, Angelina Peñaranda, Inmaculada R. Cantalapiedra, Yohannes Shiferaw, Enrique Alvarez-Lacalle.

**Software:** David Conesa.

**Supervision:** Yohannes Shiferaw.

**Validation:** Blas Echebarria, Enrique Alvarez-Lacalle.

**Visualization:** David Conesa, Enrique Alvarez-Lacalle.

**Writing – original draft:** David Conesa, Blas Echebarria, Enrique Alvarez-Lacalle.

**Writing – review & editing:** David Conesa, Angelina Peñaranda, Inmaculada R. Cantalapiedra, Yohannes Shiferaw, Enrique Alvarez-Lacalle.

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
