## [Decision Letter · Decision Letter 0]

26 Dec 2019

Dear Dr Alvarez-Lacalle,

Thank you very much for submitting your manuscript 'Two-variable nullcline analysis of ionic general equilibrium predicts calcium homeostasis in ventricular myocytes' for review by PLOS Computational Biology. Your manuscript has been fully evaluated by the PLOS Computational Biology editorial team and also by three independent peer reviewers. The reviewers appreciated the attention to an important problem, the originality of your approach and the significance of your findings. But they also asked some important questions and recommended some important revisions that I think you can address with a  thorough revision. While your manuscript cannot be accepted in its present form, we are willing to consider a revised version in which the issues raised by the reviewers have been adequately addressed. We cannot, of course, promise publication at that time.

Sincerely,

Andrew D. McCulloch, Ph.D.

Associate Editor

PLOS Computational Biology

Daniel Beard

Deputy Editor

PLOS Computational Biology

[LINK]

Reviewer's Responses to Questions

**Comments to the Authors:**

Reviewer #1: The authors address the important question of how to ascertain steady state calcium conditions in a ventricular myocoyte. To this aim, they derive two coupled equations that characterise calcium equilibria. In turn, they employ this new framework to explain counter-intuitive results from SERCA therapy that is used to treat heart failure. Overall, this is an interesting study worth publishing that fits well with PLoS Computational Biology. Before making a decision regarding its publication, I would like the authors to consider the following points.

- My main comment regards the computations involved in the derivation of equations (25) and (26). As far as I understand the approach, the authors need to numerically solve the underlying large system of coupled ordinary differential equations (ODEs) to evaluate the integrals in equations such as (20). In the computation, some variables are set to their steady state values due to time scale separation, hence reducing the dimensions of the ODE system. To then determine the nullclines, the authors need to run their simulations a large number of times to determine the surfaces in Figures 3 and 4. In turn, this is used to find the fixed point as an intersection of the nullclines. If this is indeed the case, I am unclear of the advantage of the approach. If essentially, I need to run a large number of simulations (although only over one beat) to determine the fixed point, why can I not run a single simulation for longer and then determine the steady state value? In fact, Figure 4 shows that these approaches are equivalent. It would be worth explaining in more detail why computing the nullclines (which has to be redone every time parameter values are changed) is superior to running a single long simulation. For this, I also do not see how the time scale separation gives any advantage for the proposed framework as it could equally well be applied to the single long simulation.

- The authors balance the fluxes during one beat, which is then used to determine the steady state calcium concentrations. Another way of looking at this is to write down a map that maps the calcium concentration at the beginning of a beat to that at the end of the beat and then look for fixed points of this map. If this is indeed a sensible interpretation of the authors' work, I strongly recommend to put their results into context. Maps have a long tradition in cardiac modelling as a means to reduce the high dimensionality of cardiac models. The authors should expand on this history and also highlight how their map is different from established approaches (or how it relates to them). It is also worth mentioning that a similar idea was used in Huertas, M. A., Smith, G. D., & Gyorke, S. (2010). Ca2+ alternans in a cardiac myocyte model that uses moment equations to represent heterogeneous junctional SR Ca2+. Biophysical Journal, 99(2), 377–387. To reduce the dimensionality of calcium dynamics, a recent study proposes the use of the Master stability function on a piecewise linear version of a well-established model of calcium cycling in ventricular mycoytes: Veasy, J., Lai, Y. M., Coombes, S., & Thul, R. (2019). Complex patterns of subcellular cardiac alternans. Journal of Theoretical Biology, 478, 102–114.

- The authors note that a particular advantage of their model is the explicit representation of buffer dynamics and not the use of the fast buffer approximation. As they point out, "The reason is that fast buffering approximation leads to a loss of mass in any type of propagation algorithm we have considered." In the limit of perfect time scale separation, i.e. infinitely rapid buffering, fast buffering is exact. While it is true that there is no ODE for the buffers, the algebraic relation following from the fast buffer approximation allows us to recover the buffered concentrations. There does not seem to be a "loss of mass" here. The question is then more if the assumptions of the fast buffering approximation are satisfied. Of course, if that is not true, then using it is misleading and might contribute to what the authors call "loss of mass". It is also unclear to me if their arguments are based entirely on a numerical implementation (as the authors make reference to the Euler scheme) or whether they believe it is a structural problem for all numerical algorithms. If it is the former, I suggest to change the integrator. Also, I think it is a misuse of terminology to call the authors' model mass conserving. The model clearly does not conserve overall mass since it neglects dynamics of the extracellular calcium concentration, which is indeed clamped. At the end of the results section, the authors state "The reason is that free calcium concentrations and total calcium concentrations can be related one-to-one, in a general equilibrium framework, using the fast-buffering approximation." Does this imply that the authors used the fast buffer approximation after all?

- One way of reducing the dimensionality of the model is by splitting variables into fast and slow dynamics and then express the fast variables as a function of the slow variables. Could the authors show explicitly from numerical simulations that this time scale separation holds and that they obtain the same results as with the full model, i.e. without using algebraic equations for the fast variables?

Minor comments:

- In the author summary, "2+" needs to be superscript in "Ca2+"

- l. 79: "is not a cube with". Do the authors mean that a CRU is a rectangular cuboid?

- l.157: Formally, the buffer contributions are subtracted, not added to the calcium equations.

- Equation (12) misses a bracket in the denominator.

- l.267: Why is the unit of J_LCC mol/s and not M/s?

- Could the authors provide a source for the large diffusion coefficient within a z-plane? A cytosolic diffusion coefficient of 1.2 µm^2/ms (i.e. 1200 µm^2/s) appears really large (with a similar argument for the SR).

Reviewer #2: This study from Conesa et al uses an exciting and original approach to predict and explain the homeostatic equilibrium in cardiac calcium handling. This approach is potentially very powerful, with the ability to provide both substantial predictive ability as well as mechanistic explanations for complex and often counter-intuitive observations

I have no major concerns with the study itself, which is in general well and clearly described, and certainly uses suitable approaches for the objectives. However, I do have some comments regarding the presentation and structuring of the paper.

Structure: I feel that different parts of this paper are placed in the wrong locations. The description of the general equilibrium approach I feel would be better suited to the methods, as this explains the framework underlying the paper – the validation of this method can then be the first part of the results. Moreover, the majority of the discussion reads more like results – presenting the applications of the validated model to explain the example of both intuitive and counter-intuitive responses to SERCA upregulation – including 2 results figures both presented within this discussion. I would suggest moving all of this to the results section (the final, summary figure should remain in the discussion).

Furthermore, this does mean the discussion lacks some of the content which should be present. The implications for future research, in particular in context of the explanation of the SERCA gene therapy study failure, should be expanded on. Similarly, the limitations should be clearly described. There is one limitation which, while certainly not reducing the value of this study, does need to be discussed: The clamped AP; I completely understand why this was performed and that the approach would be significantly more challenging if this were not the case, but it does need to be clearly described as a limitation. In particular, changes to LCC and NCX will directly impact the AP, and changes to the CaT and its impact on LCC and NCX will also affect the AP; these AP affects may then result in further changes to the dynamics and affect the homeostatic equilibrium, as it is under different conditions. This non-linear interaction is a key component of long-term cardiac dynamics, and should therefore be explicitly discussed.

Regarding the summary figure. I am happy with this as it is, but do feel it could be further improved by also having the pathway to increased transients on there. I do understand that this is more trivial, but it could really help understand the differences between these two opposing outcomes for the same input changes.

As a smaller comment, I think the mechanism of CICR should be introduced much earlier than its current introduction ~ ln 124. Some arguments in the intro (such as the explanation of effect on SR release) would be better supported by this basic mechanism having already been described

Minor text suggestions:

1. Please rephrase all “anti-intuitive” with “counterintuitive” (which is used in some cases).

2. Ln 12: maybe increasing the heart rate, rather than rhythm, would be clearer? A rhythm is regarding regularity rather than rate.

3. Ln 22: “Species dependence is not limited to the release of calcium: its reuptake into the SR …” may be a better way of phrasing this sentence.

4. Ln 33: dysregulations -> dysregulation.

5. Ln 37: “strategies as gene therapy” -> “strategies such as gene therapy”. This minor error is repeated a few times: I have noted the ones I have noticed but please keep a look out for further examples.

6. Ln 64 “seem clear” -> “seems clear”

7. Ln 79. The “is not a cube” statement is a little confusing, especially given that what follows does not describe what it actually is if not a cube?

8. Ln 91: “is sensible to the calcium gradient” should be “sensitive”. This error is also made a few times within the MS, and please do not assume I have noticed every instance.

9. Ln 258: typo in equilibrium (no “b”!)

10. Ln 263: CaRU has already been defined?

11. Ln 276: “Being this the case” -> “This being the case”

12. Ln 334: Please scale back a little on the description of the fit as being “perfect”

13. ln 337: Sensible -> sensitive error

14. Ln 362: “such as macroeconomics”

15. Ln 472 “behaviour such as discordant”

16. Ln 499: “clearly and in-silico failure” needs revising.

Reviewer #3: Conesa et al have investigated calcium homeostasis in ventricular myocytes using a computational model and mathematical analysis. In general, the article was interesting, and I liked that they used economic models/approaches.

I have the following questions

• Equations for JLCC and JNCX do not contain csr. Page 10, equation 25 (and Figure 4), why is it dependent on csr?

• Page 7, “We use the expression given in [26] for the properties of LCC in rabbit.” However, the model in [26] has 7 states. Please provide more details on your 5-state model. Is the model validated based on experimental data?

• Similarly, the RyR model has only one open state whereas the RyR model in [26] has two open states.

• Typical Ca_SR is 700 ~1000uM. In this study, Ca_SR is too low (<100 uM). It should be at least >500uM.

• The organization of the paper could be changed to put the sections in a better order. For example, some materials from the “Discussion” section can be moved to the “Result” section. The “Method” section can be simplified.

• Page 2, line 12, it stated “In most animal species…” I wonder if there is any animal/mammal that the amount of blood pumped at each beat doesn't increase with beat rate?

• Page 2, line 47, there is a typo “calcium”

• Figure 1, panel 2, it would be good to depict the subsarcolemmal space inside the cell too.

• Figure 1, panel 3, x-axis values are missing.

• Figure 2, all the panels don’t have x-axis values.

• Page 4, line 86, is “attached-to-buffers” same as “bound”? If yes, it would be good to be consistent and not to use different terms for the same thing.

• Page 5, from line 134 to 152, could they cite one or two articles?

• Page 9, equation 22, does inside the parenthesis (phi_i) refer to the variables of the ith CRU?

• Figure 3, lower panel, in the 3D figure, what is “fmol”?

• Figure 4, right panel, it would good to label the curves as the f- and g-nullclines (similar to Fig S1)

• Figure 5, on 3D plots, what is “fmol”?

• Figure 5, for 3D plots, could they zoom in? it is hard to see what is going on there.

• Figure 6, for x-axis, could they use the notations similar to their Supplemental Material’s figures (Fig S2 and S3)? In general, I found the Supplemental Material much easier to follow.

• Figure 7, there is a typo in the “SERCA Therapy” box.

• It is not easy to follow figure 7.

**Have all data underlying the figures and results presented in the manuscript been provided?**

Reviewer #1: Yes

Reviewer #2: Yes

Reviewer #3: None

PLOS authors have the option to publish the peer review history of their article (what does this mean?). If published, this will include your full peer review and any attached files.

Reviewer #1: No

Reviewer #2: No

Reviewer #3: No

---

## [Decision Letter · Decision Letter 1]

5 May 2020

Dear Dr. Alvarez-Lacalle,

We are pleased to inform you that your manuscript 'Two-variable nullcline analysis of ionic general equilibrium predicts calcium homeostasis in ventricular myocytes' has been provisionally accepted for publication in PLOS Computational Biology.

Best regards,

Andrew D. McCulloch, Ph.D.

Associate Editor

PLOS Computational Biology

Daniel Beard

Deputy Editor

PLOS Computational Biology

Reviewer's Responses to Questions

**Comments to the Authors:**

Reviewer #1: All of my comments have been answered satisfactorily. I now recommend the manuscript for publication.

Reviewer #2: The authors have satisfactorily addressed all of my comments. I congratulate them on a novel and interesting paper.

Reviewer #3: No further questions.

**Have all data underlying the figures and results presented in the manuscript been provided?**

Reviewer #1: Yes

Reviewer #2: Yes

Reviewer #3: None

PLOS authors have the option to publish the peer review history of their article (what does this mean?). If published, this will include your full peer review and any attached files.

Reviewer #1: No

Reviewer #2: No

Reviewer #3: No

---

## [Editor Report · Acceptance letter]

29 May 2020

PCOMPBIOL-D-19-02078R1 

Two-variable nullcline analysis of ionic general equilibrium predicts calcium homeostasis in ventricular myocytes

Dear Dr Alvarez-Lacalle,

I am pleased to inform you that your manuscript has been formally accepted for publication in PLOS Computational Biology. Your manuscript is now with our production department and you will be notified of the publication date in due course.

With kind regards,

Laura Mallard
